

# Categorization of tweets for damages: infrastructure and human damage assessment using fine-tuned BERT model

Muhammad Shahid Iqbal Malik[1], Muhammad Zeeshan Younas[2], Mona Mamdouh Jamjoom[3] and Dmitry I. Ignatov[1]

[1] Department of Computer Science, National Research University Higher School of Economics, Moscow, Russia

[2] Department of Computer Science, Capital University of Science and Technology, Islamabad, Pakistan

[3] Department of Computer Sciences, College of Computer and Information Sciences, Princess Nourah bint Abdulrahman University, Riyadh, Saudi Arabia

## ABSTRACT

Identification of infrastructure and human damage assessment tweets is beneficial to disaster management organizations as well as victims during a disaster. Most of the prior works focused on the detection of informative/situational tweets, and infrastructure damage, only one focused on human damage. This study presents a novel approach for detecting damage assessment tweets involving infrastructure and human damages. We investigated the potential of the Bidirectional Encoder Representations from Transformer (BERT) model to learn universal contextualized representations targeting to demonstrate its effectiveness for binary and multi-class classification of disaster damage assessment tweets. The objective is to exploit a pre-trained BERT as a transfer learning mechanism after fine-tuning important hyper-parameters on the CrisisMMD dataset containing seven disasters. The effectiveness of fine-tuned BERT is compared with five benchmarks and nine comparable models by conducting exhaustive experiments. The findings show that the fine-tuned BERT outperformed all benchmarks and comparable models and achieved state-of-the-art performance by demonstrating up to 95.12% macro-f1-score, and 88% macro-f1-score for binary and multi-class classification. Specifically, the improvement in the classification of human damage is promising.

## INTRODUCTION

Disasters may cause monolithic destruction and sometimes create uncontrollable and unpredictable situations. Natural disasters are caused by natural phenomena like wildfires, floods, *etc.*, and their intensities affect the proportion of lives, the environment, and the economy of an area (*Koshy & Elango, 2023*). During disasters, public and private organizations rely on critical and timely information to set up required operations for

Corresponding author
Muhammad Shahid
Iqbal Malik, mumalik@hse.ru,
shahid.msimalik@gmail.com

helping affected people. The development of web technologies enables people to use social media platforms like Twitter, *etc.*, to exchange their views and recent happenings in their suburbs. Twitter is one of the most popular and widely used platforms that facilitate tweets up to 280 characters at maximum. More than 486 million users are active on Twitter according to recent statistics and more than 1.4 trillion tweets are posted annually (*Madichetty, Muthukumarasamy & Jayadev, 2021*).

For humanitarian authorities, the assessment of disaster damage is one of the critical steps to get the real situation and seriousness of damage so that services accordingly can be provided. It is a common practice during the disaster and in the aftermath that people place massive messages on Twitter related to situational information (*Madichetty, 2020*; *Madichetty & Sridevi, 2020*; *Rudra et al., 2018*). Therefore, the identification of damage assessment of social media posts like tweets, *etc.* is one of the important aspects. Several studies (*Cresci et al., 2015*; *Nguyen et al., 2017*; *Priya et al., 2018*; *Rudra et al., 2018*) addressed this task of damage assessment. *Cresci et al. (2015)* focused on the Italian tweets by handling damage assessment of buildings and infrastructure only and did not address human damage assessment. Likewise, *Rudra et al. (2018)* proposed a model for situational tweet identification and their summarization for English and Hindi tweets but did not address human damage assessment. *Nguyen et al. (2017)* also missed human damage assessment and important information from textual data and focused only on buildings and infrastructure damage using image data. *Priya et al. (2018)* developed a query-based information retrieval method for infrastructural damage assessment but did not address human damage. Moreover, few studies proposed approaches for informational *vs* non-informational tweet identification like the majority voting approach (*Krishna, Srinivas & Prasad Reddy, 2022*), and multi-model approach using image and text data (*Koshy & Elango, 2023*) but did not address human damage assessment. According to our knowledge, only one study, *Madichetty & Sridevi (2021)*, focused on damage assessment for infrastructural and human damages from tweets and used a lexicon and frequency-based approach (hand-crafted) with traditional machine learning (ML) models. However, the related studies missed the utilization of language models for infrastructure and human damage assessment from tweets, in contrast, they mainly used hand-crafted features. Moreover, the performance achieved by the baseline (*Madichetty & Sridevi, 2021*) is not promising. To fulfill this gap, two objectives are devised in this study:

1. To design an automated damage assessment approach to identify infrastructural and human damages from tweet data using a state-of-the-art language model.
2. To demonstrate the effectiveness and outperformance of the proposed automated approach in comparison with benchmarks through experimental results.

To capture the actual context of the language used to describe human and infrastructural damages in tweets, a state-of-the-art language model is required. Fine-tuning the BERT model has demonstrated robust performance in similar natural language processing (NLP) tasks (*Malik, Cheema & Ignatov, 2023*; *Malik, Imran & Mamdouh, 2023*). Therefore, we are interested in utilizing the architecture of the BERT language model with fine-tuning. The novelty of the study is threefold: First, the proposed framework is based on an automated feature generation model in contrast to hand-crafted features. Second, the fine-tuning of

the BERT transformer model is performed not only for damage tweet identification but also for human and infrastructure damage detection. Third, the proposed framework delivered benchmark performance and outperformed the standard baselines and comparable models. The proposed study contributes in the following ways:

1. According to our knowledge, the fine-tuning of BERT is used first time for the identification of infrastructure and human damage assessment tweets.
2. The utilization of contextual semantic embeddings helped us to handle the ambiguity and complexity issues of the seven disasters of the CrisisMMD dataset.
3. The fine-tuning of BERT for binary and multi-class classification on seven disasters showcases substantial improvement in performance as compared to five benchmarks and nine comparable models.
4. The optimization of hyperparameters is performed for the BERT model to handle the overfitting and catastrophic forgetting issues and to obtain benchmark performance.
5. The improvement achieved by the proposed framework is proved to be statistically significant and verified by the Wilcoxon signed-ranked test method.
6. An extensive set of experiments demonstrates that fine-tuned BERT achieved up to 95.12% macro f1-score for binary classification and 88% macro f1-score for multi-class classification.

The remaining part of the article is organized as follows: related work is described in 'Related Work' followed by 'Framework Methodology', in which the proposed methodology is described with fine-tuning BERT details. 'Experimental Results and Analysis' presents the dataset description, and experimental setup, and discusses results in detail. 'Conclusion' concludes the research work and presents future directions.

## RELATED WORK

In this section, we review the literature that addresses the issue of assessment of social media posts for various damages and disaster detection and summarization approaches.

In 2014, an automated classification approach for informative tweets was designed (*Imran et al., 2014*). The authors named their model "Artificial Intelligence for Disaster Response (AIDR)", tested it on the Pakistani Earthquake dataset, and achieved 90% area under curve (AUC). Later, *Cresci et al. (2015)* proposed an infrastructure damage assessment detection model for Italian tweets. Their method used SVM with a variety of linguistic features and they claimed that their approach was the first to be tested on non-English data. Then, *Nguyen et al. (2017)* utilized the VGG-16 vision model and bags of visual words to build a multi-class classification framework. Their model was tested on several disaster datasets and VGG-16 outperformed the bags of visual words approach. Likewise, the study *Rudra et al. (2018)* proposed a two-step methodology to extract situational information and then summarization for disaster tweets in English and Hindi languages. The low-level lexical and syntactic features with SVM classifier are explored. They claimed that non-English tweets are explored first time.

In 2019, the Domain-Adversarial Neural Network (DANN) model was used with VGG-19 to identify the damages from image data (*Li et al., 2019*). They claimed that

their approach demonstrated significant performance but they did not address human damage identification. Then, several ML and deep learning (DL) models are explored with the Term Frequency-Inverse Document Frequency (TF-IDF) model to classify the informative microblog posts into multi-categories (*Kumar, Singh & Saumya, 2019*). The authors compared the performances of ML and DL models and the best results are reported. Another image data-based study is conducted by *Imran et al. (2020)*. The proposed approach classified the tweets into damage or non-damage and then categorized the tweets based on severity like severe *vs* mild *vs* non-damage but they did not address human damage identification. Later, an information retrieval approach is utilized to assess infrastructure damage tweets (*Priya et al., 2020*). They developed topic topic-aligned query expansion method and evaluated it on several disaster datasets. Similarly, *Alam, Ofli & Imran (2020)* analyzed the situational characteristics of three hurricane disasters and developed a multi-class classification model using random forest. Their findings revealed that both text and image data contain important information.

In 2021, *Alam et al. (2021)* attempted to combine various crisis datasets to facilitate binary and multi-class classification. The authors used the convolutional neural network (CNN) model with FastText embeddings to explore the impacts of these approaches and conclusions are drawn. As noted earlier, there is only one study, *Madichetty & Sridevi (2021)*, that addressed the issue of infrastructure and human damage collectively from tweets. We chose this study as one of the baselines. The authors used lexicons and TF-IDF features with six ML models and revealed that their framework outperformed the baselines. Later, a majority voting-based approach is presented in *Krishna, Srinivas & Prasad Reddy (2022)* to identify only informative tweets using word2vec, TF-IDF, and the Glove model. Their model showed significant performance but they did not handle infrastructure and human damages. A real-time damage assessment of tweets using image data is performed by *Imran et al. (2022)*. The authors developed a system using computer vision models and determine the severity of damages. Later, *Koshy & Elango (2023)* derived an approach for informative tweet identification using the Robustly Optimized BERT (RoBERTa) model with bidirectional long short term memory (bi-LSTM) on textual and image data. Their results demonstrated the importance of their binary model.

Recently, another DL-based approach is presented by *Paul, Sahoo & Balabantaray (2023)* to classify disaster tweets into binary and multi-class categories. The authors used CNN, GRU, and SkipCNN models, and their model showed significant improvement. Then, the authors *Alam et al. (2023)* proposed a multi-task learning framework using image data. They released a dataset and conducted binary and multi-class classification tasks using DL models. Their model showed significant performance. Likewise, *Lv, Wang & Shao (2023)* built an auto-encoder-based model for classifying crisis-related tweets. Textual and image data are used to test the model and their model outperformed the benchmark. Then, a disaster summarization method was proposed by *Garg, Chakraborty & Dandapat (2023)* using the ontology technique and they tested their model on twelve disaster datasets. Their model outperformed the baselines. A recent multi-class classification approach for disaster tweets is presented *Asinthara, Jayan & Jacob (2023)* and they used TF-IDF and word2vec features with SVM and Bi-LSTM models. Thei r results show that with the SVM model, the

performance is significant. Likewise, the identification of high-priority tweets is performed by *Arathi & Sasikala (2023)*. Their model used Glove embeddings and metadata features with the random forest model and achieved 91% accuracy and 94% f1-score.

More recently, *Dasari, Gorla & Prasad Reddy (2023)* built a classification system for the detection of informative tweets. A stacking ensemble model is proposed and is used with TF-IDF, word2vec, and Golve feature models. Their model demonstrated better performance than baselines. For informative tweet categorization, the latest approach used ontology infused DL model (*Giri & Deepak, 2023*). They tested their approach on the image and textual dataset and claimed that their model presented benchmark performance. Likewise, *Madichetty & Madisetty (2023)* developed a detection pipeline for multi-modal disaster tweets. They utilized RoBERTa, and VGG-16 models for feature extraction and combined their output using a fusion method. Their model outperformed the baselines. Another study handled the classification of disaster tweets by exploring bag-of-words and several ML models (*Iparraguirre-Villanueva et al., 2023*). The highest performance achieved is 87% accuracy.

The summary of prior approaches related to damage assessment is presented in Table 1. In contrast, some latest approaches focused on the issue of identification of emergency messages relevant to first responders like the study *Powers et al. (2023)* proposed a framework to identify emergency tweets and then categorized them according to relevancy and urgency level. The authors used BERT and XLNet transformers with the CNN model and their model showed promising performance. We found the following limitations in the literature regarding damage assessment of disaster tweets:

- **Lack of human damage assessment:** To the best of our knowledge, only one study has focused on the assessment of human damage as well as infrastructure damage.
- **Effective feature engineering**: Most of the studies used linguistic, syntactic, and frequency-based features (hand-crafted), but missed language models and their fine-tuning.

## FRAMEWORK METHODOLOGY

In this section, the detail of the proposed framework is described. At the first level, the framework performs the detection of damaged or not-damaged tweets. At the second level, it further classifies the damage tweets into infrastructure damage or human damage. The pipeline of the proposed framework is presented in Fig. 1. The CrisisMMD dataset is preprocessed by applying several steps (Section 'Data Pre-processing'), and then the dataset is split into an 80–20 ratio (80% training and 20% testing). After that, the dataset is transformed into a specified format so that it can be used as an input to transformer mode. Then the fine-tuning of BERT is performed using the grid search technique for hyper-parameters optimization. The results of the comparable models and state-of-the-art baselines are generated and compared with the fine-tuned BERT model. In the end, the conclusions are drawn. The pseudo-code of the proposed methodology is presented in the Fig. 2.

**Table 1  Summary of literature for informative tweets, infrastructure, and human damage identification.**

| Ref | Tasks | | | Features | Supervised models |
|---|---|---|---|---|---|
| | IN/DM | IF DM | H DM | | |
| *Imran et al. (2014)* | ✓ | | | Customized model | AIDR model |
| *Cresci et al. (2015)* | | ✓ | | Linguistic and *ad-hoc* features | SVM |
| *Nguyen et al. (2017)* | ✓ | ✓ | | Bag of visual words, VGG-16 | CNN |
| *Rudra et al. (2018)* | ✓ | | | Bag of words, low-level lexical, syntactic features | SVM |
| *Kumar, Singh & Saumya (2019)* | ✓ | | | TF-IDF, Glove | RF, SVM, KNN, NB, CNN, LSTM, GRU |
| *Li et al. (2019)* | | ✓ | | VGG-19 | Domain-Adversarial Neural Network |
| *Priya et al. (2020)* | | ✓ | | Latent Dirichlet allocation, Part of speech | Information retrieval |
| *Alam, Ofli & Imran (2020)* | | ✓ | | Bag of words, sentiment, Spatial features | RF |
| *Imran et al. (2020)* | ✓ | ✓ | | Image features | DNN |
| *Alam et al. (2021)* | ✓ | | | FastText | CNN |
| *Madichetty & Sridevi (2021)* | ✓ | ✓ | ✓ | Low-level lexical, syntactic, and frequency features | SVM, RF, GB, AB, Bagging |
| *Krishna, Srinivas & Prasad Reddy (2022)* | ✓ | | | TF-IDF, word2vec, Glove | Majority-voting ensemble model |
| *Imran et al. (2022)* | ✓ | | | —— | Computer vision models |
| *Koshy & Elango (2023)* | ✓ | | | Roberta, biLSTM | biLSTM |
| *Paul, Sahoo & Balabantaray (2023)* | ✓ | ✓ | | Word embeddings, bag of words | CNN, GRU, SkipCNN |
| *Alam et al. (2023)* | ✓ | ✓ | | ResNet18, VGG16 | Deep learning models |
| *Lv, Wang & Shao (2023)* | ✓ | | | ResNet50, Word embeddings | Bi-GRU |
| *Garg, Chakraborty & Dandapat (2023)* | ✓ | ✓ | | —— | Ontology-based approach |
| *Asinthara, Jayan & Jacob (2023)* | ✓ | ✓ | | TF-IDF, word2vec | SVM, bi-LSTM |
| *Arathi & Sasikala (2023)* | ✓ | | | Glove model, metadata features | RF |
| *Dasari, Gorla & Prasad Reddy (2023)* | ✓ | | | TF-IDF, word2vec, Glove | Stacking ensemble model |
| *Giri & Deepak (2023)* | ✓ | | | Image features, word embeddings | Ontology infused DL model |
| *Madichetty & Madisetty (2023)* | ✓ | | | RoBERTa, VGG-16 | Fusion method |
| *Iparraguirre-Villanueva et al. (2023)* | ✓ | | | Bag-of-words | LR, KNN, DT, RF, NB |
| *Proposed | ✓ | ✓ | ✓ | Fine-tuning of BERT, TF-IDF, word2vec | BERT classifier, RF, SVM |

**Notes.**

IN, Informative; DM, Damage; IF, Infrastructure; H, Human; SVM, Support Vector Machine; RF, Random Forest; GB, Gradient Boosting; AB, AdaBoost; LSTM, Long short-term Memory Network; KNN, K-nearest neighbor; NB, Naïve Bayes; CNN, Convolutional Neural Network; GRU, Gated Recurrent Unit; AIDR, Artificial Intelligence for Disaster Reponses; DNN, Deep Neural Network; LR, Logistic Regression; DT, Decision Tree.

## Data pre-processing

The following pre-processing steps are employed before providing data to the fine-tuning process and extraction of TF-IDF, and word2vec features.

1. Removal of hashtags, HTML tags, mentions, punctuations, URLs, and numbers.
2. Conversion of tweets to lowercase.
3. Replacement of the emoji/emoticons with their corresponding text.

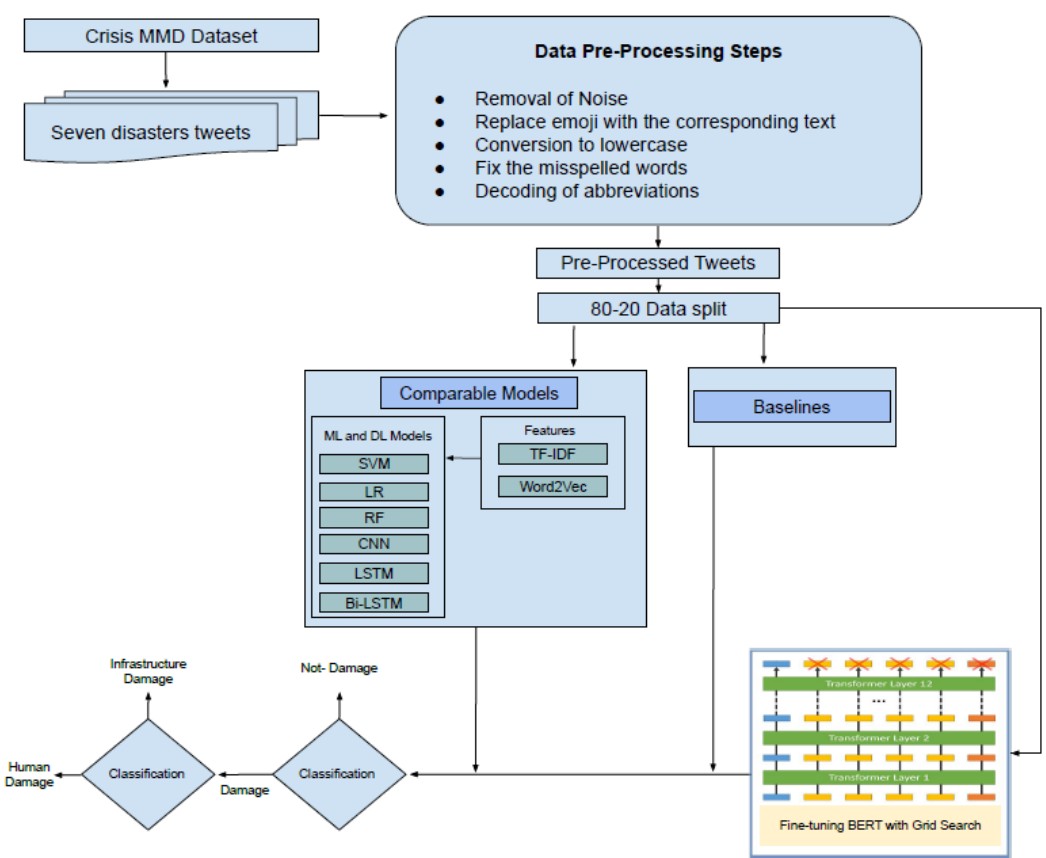

**Figure 1** Disaster damage tweets identification framework.

4. Fixing the issue of misspelled words.
5. Decoding of abbreviations (thnx, thx, btw, pls, plz *etc.*).
6. Removal of stop words (Only for TF-IDF and word2vec).

## Fine-tunning BERT

The BERT model was introduced by *Devlin et al. (2018)* at Google Lab and it has proven its significance for a variety of text-mining tasks in several application domains (*Malik, Imran & Mamdouh, 2023*). The benefits of BERT include faster development, automated feature generation, reduced data requirements, and improved performance. It has two architectures and we are interested in fine-tuning the pre-trained BERT model for damage assessment tweet identification task for binary as well as multi-class classification. The BERT model is pre-trained on a large corpus of English data in a self-supervised fashion and uses the context of language in both directions. Furthermore, BERT was pre-trained on next-sentence prediction and masked language modeling objectives.

To fine-tune the BERT base uncased model (https://huggingface.co/bert-base-uncased), some important steps are required. After applying the above-mentioned pre-processing steps, data transformation and training classifier steps are executed.

```
Algorithm 1: : Disaster Damage Tweets Identification (Dataset D)
// The dataset D = {Diaster1, Diaster2, ……………Diaster 7}, the dataset consists of seven diasters
1: procedure  Disaster Damage Tweets Identification (Dataset D)
2:      D-P ← Pre-Processing (D);                    // dataset pre-processing
3:      for i = 1 to 7                               // for loop to consider each disaster data one by one
4:          Disasterᵢ ← Select (D-P)                 // select each disaster and proceed further
5:          Proposed-Method (Disasterᵢ);             // classification of damage tweets using fine-tunning
6:      end for
7: end procedure
1: procedure Pre-Processing (D)
2:      D1  ← Cleaning (D);                          // Perform cleaning (remove noise)
3:      D2  ← Lower-Case (D1);                       // Lower-case conversion
4:      D3  ← Replace-Emoji (D2);                    // Replace emojis with the corresponding text
5:      D4  ← Decode-Abbreviation (D3);              // Decode abbreviations
6:      return D4;
5: end procedure
1: procedure Proposed-Method (Disaster Da)          // classification using fine-tuning of BERT
2:      Da ← BERT-Tokenizer (Da);                    // Apply BERT Tokenizer
3:      Model ←  fine-tunning (Da, BERT, 80-20)      // Fine-tuning of BERT by applying 80-20 data split
4:      confusion-matrix ← generate-results (Model);
5:      accuracy ← compute-accuracy (confusion-matrix);
6:      precision ← compute-precision (confusion-matrix);
7:      recall ← compute-recall (confusion-matrix);
8:      f1-score ← compute-f1(confusion-matrix);
9: end procedure
```

**Figure 2** **Pseudo-code of the proposed tweets identification framework (Algorithm 1).**

**Data transformation:** To transform data into a predefined format, we tokenized each tweet text into N tokens by the uncased BERT tokenizer. The list of N tokens is modified by adding the [CLS] token at the beginning and the [SEP] token at the end. Then an input representation based on pre-trained BERT vocabulary is generated for each token. The [CLS] token generates word embeddings that derive the input data for classification.

We have chosen 64 and 128 sequence lengths because a maximum of 280 characters are allowed in a tweet and 128 sequence length is enough to handle the most lengthy tweets. Therefore, all tweets are padded up to the length of 64 and 128. After that, attention masks are added to locate real and padded tokens. The vector output of attention masks is then fed to the BERT model and fine-tuning step is performed.

**BERT classifier training:** There are seven disasters in the CrisisMMD dataset. For training and validation of BERT classifiers, we split each disaster into 80-20 ratios using the stratified sampling approach. After that, we took 80% data from each disaster and combined them to make the training dataset. The remaining 20% of data from each disaster is used for testing the BERT classifier on that specific disaster data. Furthermore, the combined 80% data is further divided into a 90-10 split, in which 90% is used for training and 10% is used for validation. We utilize the BERT base model which contains 12 transfer layers, 12 attention heads, and 768 hidden layers. All entities (class labels, token ids, and attention masks) are combined into one set.

For the classification of damage tweets, we attach the outputs of BERT (after fine-tunning) with an additional layer consisting of Softmax classifier as shown in Fig. 3. We denote Ti as the final hidden vector for *ith* token and *h* as a final hidden vector of [CLS]

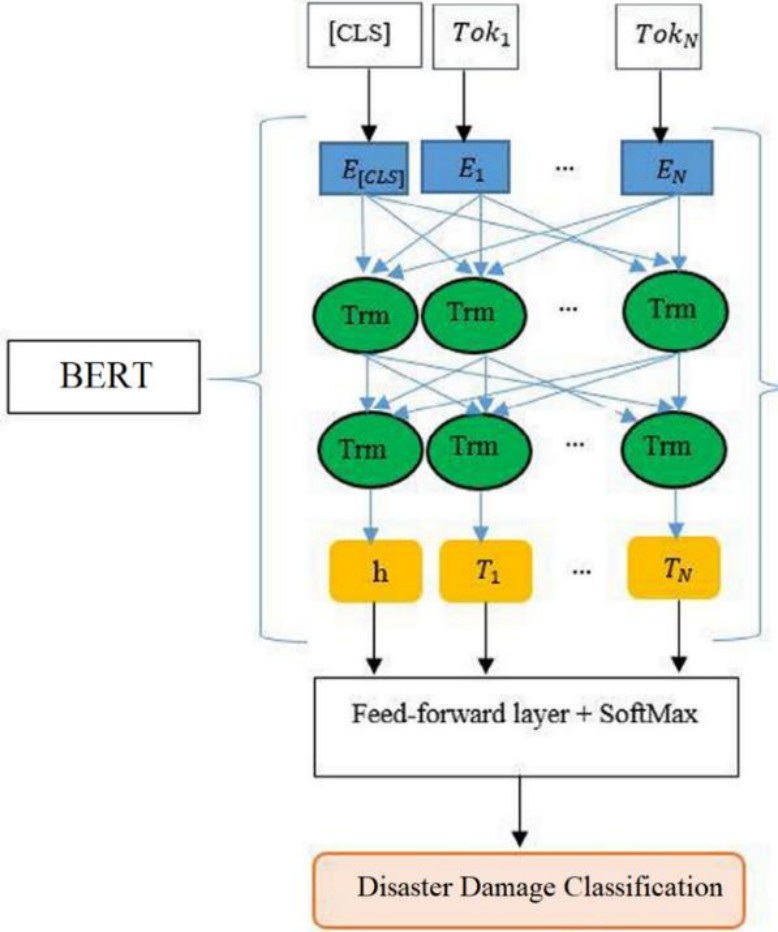

**Figure 3   Architecture of proposed fine-tuned BERT for damage classification.**

token. This h and all Ti vectors are fed to the feed-forward layer containing the Softmax classifier to get the predicted labels. For fine-tuning, the hyperparameters are chosen by using the guidelines of prior studies *Devlin et al. (2018)*. The batch sizes of 8, 16, and 32 are explored with sequence lengths of 64 and 128 but we added the best results (Section 'Fine-tunning of BERT and comparison with baselines (damage *vs* non-damage)') which are obtained by using 32 batch size. For each epoch, we use the optimizer method to update the parameters, and the output of the trained model is evaluated on the validation set by calculating validation loss, validation accuracy, and validation f1-score. Overall, six BERT classifiers are fine-tuned and we use the Google Colab platform for experimental setup. The detail of hyperparameters is presented in Table 2.

**Catastrophic forgetting:** The literature demonstrates that while fine-tuning a language model to learn new knowledge, previously learned knowledge may be lost because we unfreeze weights. Researchers *Sun et al. (2019)* call it catastrophic forgetting in transfer learning and every transformer model is prone to this effect. A range of learning rates is explored to get insights and examine the effects of learning rate on catastrophic forgetting

**Table 2  Hyperparameters for fine-tuning BERT for the CrisisMMD dataset.**

| Sequence length | Epochs | Batch size | Learning rate | Epsilon |
|---|---|---|---|---|
| 64 | 4 | 8, 16, 32 | 2e−5 | 1e−8 |
| 128 | 4 | 8, 16, 32 | 2e−5 | 1e−8 |

while fine-tuning BERT. The learning rates are 1e−4, 1e−5, 2e−5, 3e−4, 3e−5, 5e−5 respectively. In the training process, all layers of BERT are unlocked so that weights can be updated in all layers during the fine-tuning cycle. After repeating the training process several times and careful monitoring, it allows us to select our starting learning rate. We concluded that fine-tuning with higher learning rates (3e−4, 3e−5, and 5e−5) could lead to convergence failure. The best performance was observed with a learning rate of 2e−5 and this lessens the risk of catastrophic forgetting in fine-tuning.

**Overfitting:** How to choose the appropriate number of epochs for fine-tuning? It is a common issue for fine-tuning the transformers and deep learning models. So many epochs result in overfitting problems whereas very few may cause under-fitting. There are several methods for selecting an appropriate number of epochs, one can start with a large number of epochs and can stop the training process when no improvement is observed on the selected metric. In this research, we use validation loss as a measure to monitor the performance of the BERT classifier. We concluded that four epochs are an appropriate number to avoid overfitting issues.

## Word2vec

Word2vec is an algorithm that is used to generate "distributed word representations" inside a dataset (*Ali, 2019*). In addition, it can generate a vector of a specific length for each word by taking a sentence as input. Word2vec has demonstrated significant performance in similar NLP tasks (*Ali & Malik, 2023*; *Hussain, Malik & Masood, 2022*; *Younas, Malik & Ignatov, 2023*). The skip-gram and continuous bag of words (CBOW) are the two algorithms supported by the word2vec model to generate word embeddings. We are interested in using the skip-gram model to generate embedding features. The skip-gram model tries to predict relevant contextual words for an input word. Window size is another parameter used to confine the number of context words in a frame and we use a window size of 100 dimensions.

## TF-IDF

TF-IDF is a statistical approach to evaluate the significance of a particular word in a large context of the document. This technique is commonly used in NLP and information retrieval (IR) tasks (*Malik, Imran & Mamdouh, 2023*). It is a weighting technique and the weight of a word in a document is proportional to its frequency of occurrence whereas it is also inversely proportional to its frequency in all documents.

**Table 3  Detail of CrisisMMD dataset.**

| Disaster | Infrastructure | Human | Non-damage | Total |
|---|---|---|---|---|
| Hurricane Irma | 440 | 168 | 3,896 | 4,504 |
| Hurricane Harvey | 402 | 238 | 3,794 | 4,434 |
| Sri Lanka Floods | 46 | 44 | 932 | 1,022 |
| California Wildfires | 168 | 200 | 1,222 | 1,590 |
| Iraq–Iran Earthquake | 30 | 139 | 428 | 597 |
| Hurricane Maria | 298 | 132 | 4,126 | 4,556 |
| Mexico Earthquake | 105 | 171 | 1,104 | 1,381 |

## EXPERIMENTAL RESULTS AND ANALYSIS

In this section, a description of the dataset and details of the experimental setup are presented. Then results are conducted and analyzed to evaluate the effectiveness of the proposed framework.

### Dataset

This study used a benchmark publicly available dataset, *i.e.,* CrisisMMD (*Alam, Ofli & Imran, 2018*), to test the effectiveness of the proposed framework. The dataset consists of information about seven natural disasters like floods, earthquakes, wildfires, hurricanes, *etc*. There are seven disaster files in the CrisisMMD dataset. Each disaster contains tweets related to a specific type of event/disaster that occurred at particular/different locations. The tweet text describes human and infrastructure damages. Originally, the tweets of the dataset had several types of class labels like displaced people, affected individuals, *etc*. As described earlier, we address damage assessment tweet identification at two levels: binary (damage *vs* non-damage) and multi-class (infrastructure damage *vs* human damage *vs* non-damage) classification. The final labels of tweets are derived as follows:

- **Infrastructure damage class:** In this class "infrastructure damage, utility damage, vehicle damage & restoration, and casualties" are combined.
- **Human damage class:** In this class "affected individuals, injured or dead people, missing, trapped or found people, displaced people, and evacuations" are combined.
- **Damage class:** In this class "infrastructure and human damage classes" are combined.
- **Non-damage class:** Tweets that describe no damage.

After the compilation of the above-mentioned categorization on seven disaster files, the final form of the dataset is described in Table 3.

### Experimental setup

We used Python language to calculate the results. Four evaluation metrics are chosen to evaluate the performance of the fine-tuned BERT, comparable models, and five baselines. The metrics are precision, recall, accuracy, and f1-score. In addition, the Wilcoxon signed rank statistical test (*Woolson, 2007*) is used to determine whether the improvements are statistically significant or not. Six state-of-the-art classifiers are chosen and comparable models are designed to compare the performance of the fined-tuned BERT model. The

classifiers are random forest (RF), logistic regression (LR), support vector machine (SVM), CNN, LSTM, and Bi-LSTM. The reason why we chose these ML and DL models is that they presented a significant performance in similar NLP and text mining tasks (*Malik et al., 2023*; *Rehan, Malik & Jamjoom, 2023*). The following comparable models are designed:

1. Word2vec+RF
2. Word2vec+LR
3. Word2vec+SVM
4. TF-IDF+RF
5. TF-IDF+LR
6. TF-IDF+SVM
7. TF-IDF+LSTM
8. TF-IDF+Bi-LSTM
9. TF-IDF+CNN

Furthermore, to compare the performance of the proposed framework with benchmark studies, we have chosen the following studies from the literature.

1. *Rudra et al. (2018)* used syntactic and low-level lexical features with an SVM model for binary and multi-class classification of damage tweets and evaluated their methodology on the CrisisMMD dataset. This is one of the baselines for comparing fine-tuned BERT performance in binary and multi-class classification tasks.
2. The authors in *Madichetty & Sridevi (2021)* used syntactic, low-level lexical, and top-frequency features with a weighted SVM classifier. Binary and multi-class classification frameworks are designed for the identification of damaged tweets.
3. *Alam, Ofli & Imran (2020)* proposed a system for damage assessment tweets identification and we chose it to compare for binary classification. They used bag-of-words features with the RF model.
4. *Kumar, Singh & Saumya (2019)* explored the impact of TF-IDF with several ML and DL models for the identification of damage assessment tweets as a binary classification. The best results are chosen for the comparison.
5. Lastly, *Alam et al. (2021)* used CNN and FastText embeddings to design a binary classification system for damage tweet identification. We compared this study with the proposed framework for binary classification.

### Fine-tunning of BERT and comparison with baselines (damage *vs* non-damage)

In this section, we performed fine-tuning of BERT for the binary classification task (damage *vs* non-damage) and then compared its performance with baselines. The results of experiments by applying sequence lengths of 64 and 128 and batch size of 32 are presented in Table 4. Although we tried eight and 16 batch sizes for these experiments, we obtained the best results with a 32 batch size so we only reported results with 32 batch size. The experiments are conducted using a learning rate of 2e−5, Epsilon of 1e−8, and four epochs. For each sequence length and epoch; training loss, validation loss, validation accuracy, validation f1-score, and training & validation times are reported.

**Table 4   Results of training and validation for several BERT classifiers (damage *vs* non-damage).**

| Sequence length | Batch size | Epochs | Learning rate | Epsilon | Training loss | Validation loss | Validity accuracy | Validating F1-score | Training time | Validation time |
|---|---|---|---|---|---|---|---|---|---|---|
| BERT-64 | 32 | 1 | 2e−5 | 1e−8 | 0.07 | 0.12 | 0.9094 | 0.8244 | 0:02:11 | 0:00:05 |
| | | 2 | 2e−5 | 1e−8 | 0.09 | 0.24 | 0.9120 | 0.8928 | 0:01:58 | 0:00:04 |
| | | 3 | 2e−5 | 1e−8 | 0.07 | 0.37 | 0.9120 | 0.8928 | 0:01:58 | 0:00:04 |
| | | 4 | 2e−5 | 1e−8 | 0.03 | 0.39 | 0.9055 | 0.8262 | 0:01:58 | 0:00:04 |
| BERT-128 | 32 | 1 | 2e−5 | 1e−8 | 0.29 | 0.23 | 0.9245 | 0.8479 | 0:03:57 | 0:00:09 |
| | | 2 | 2e−5 | 1e−8 | 0.19 | 0.25 | 0.9333 | 0.8633 | 0:03:56 | 0:00:09 |
| | | 3 | 2e−5 | 1e−8 | 0.12 | 0.29 | 0.9245 | 0.8479 | 0:03:56 | 0:00:09 |
| | | 4 | 2e−5 | 1e−8 | 0.08 | 0.33 | 0.9245 | 0.8479 | 0:03:55 | 0:00:09 |

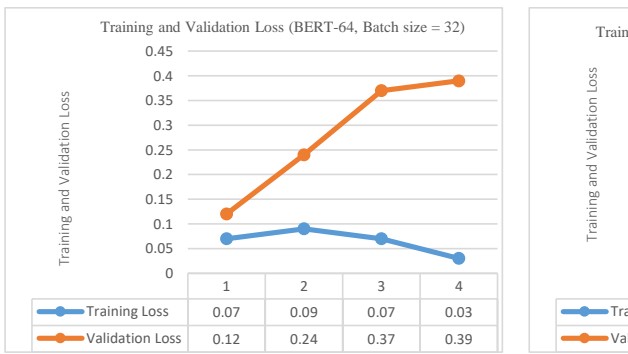 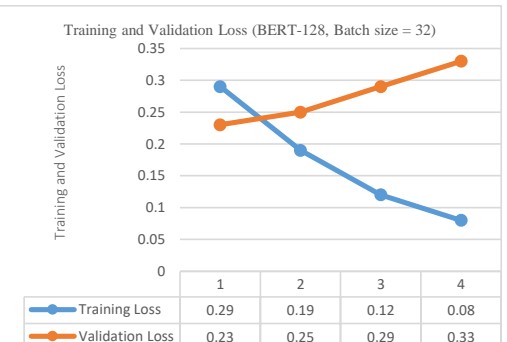

**Figure 4   Training loss and validation loss (damage *vs* non-damage) (for sequence length of 64 and batch size of 32 (left), (for sequence length of 128 and batch size of 32 (right)).**

In the first step, the BERT base model was trained and validated to classify the tweets into damage or non-damage using 64 sequence length and 32 batch size for four epochs. The training and validation loss is presented in Fig. 4 (left side). The training and validation outcomes in the form of training and validation loss, validation accuracy, and validation f1-score are presented in the upper part of Table 4. The training loss increases up to epoch 2 but then starts decreasing continuously up to epoch 4. In contrast, validation loss continuously increases from epochs 1 to epoch 4, indicating that further training could overfit the BERT model. The best values for validation accuracy and f1-score are obtained on epoch 2, *i.e.,* 91.20% and 89.28%.

In the second step, BERT was again trained and validated for four epochs using 128 sequence length and 32 batch size and results are reported in the lower part of Table 4. The training and validation loss is presented in Fig. 4 (right side) . The validation accuracy of BERT classifiers is higher than with 64 sequence length but validation f1-score is decreased. On epoch 2, the best accuracy is achieved, *i.e.,* 93.33%, and the best f1-score is 86.33%. The training loss decreases steadily and converges to 0.08 value. In contrast, validation loss increases continuously from epoch 1 to 4, indicating that further training is not useful.

**Table 5  Results of best BERT classifiers on the test part of the seven disasters (damage *vs* non-damage).**

| Disasters | Sequence length | TP | FN | TN | FP | Accuracy | Macro-precision | Macro-recall | Macro-f1-score |
|---|---|---|---|---|---|---|---|---|---|
| Iraq–Iran earthquake | 64 | 67 | 2 | 27 | 3 | 0.9495 | 0.9441 | 0.9355 | 0.9396 |
| | 128 | 68 | 1 | 27 | 3 | 0.9596 | 0.9610 | 0.9428 | 0.9512 |
| Sri Lanka floods | 64 | 140 | 11 | 14 | 0 | 0.9333 | 0.7800 | 0.9636 | 0.8401 |
| | 128 | 141 | 10 | 14 | 0 | 0.9394 | 0.7917 | 0.9669 | 0.8513 |
| Mexico earthquake | 64 | 180 | 9 | 39 | 8 | 0.9280 | 0.8850 | 0.8911 | 0.8880 |
| | 128 | 179 | 10 | 38 | 9 | 0.9195 | 0.8719 | 0.8778 | 0.8748 |
| California wildfires | 64 | 243 | 31 | 52 | 16 | 0.8626 | 0.7824 | 0.8258 | 0.8003 |
| | 128 | 242 | 32 | 49 | 19 | 0.8509 | 0.7661 | 0.8019 | 0.7812 |
| Hurricane Harvey | 64 | 631 | 52 | 77 | 34 | 0.8917 | 0.7729 | 0.8088 | 0.7889 |
| | 128 | 636 | 47 | 74 | 37 | 0.8942 | 0.7783 | 0.7989 | 0.7880 |
| Hurricane Maria | 64 | 679 | 43 | 55 | 19 | 0.9221 | 0.7670 | 0.8418 | 0.7979 |
| | 128 | 670 | 52 | 55 | 19 | 0.9108 | 0.7432 | 0.8356 | 0.7787 |
| Hurricane Irma | 64 | 658 | 31 | 76 | 35 | 0.9175 | 0.8299 | 0.8198 | 0.8247 |
| | 128 | 658 | 31 | 75 | 36 | 0.9163 | 0.8278 | 0.8153 | 0.8214 |

Hence, validation loss has demonstrated a symmetrical pattern (increasing) for both sequence lengths, and training loss is continuously decreasing.

In the third step, we tested the BERT classifiers (previously trained and validated) on the test parts of the seven disasters. For each configuration (64 and 128 sequence lengths), the BERT classifiers are tested for each epoch, but we only reported the best results against each sequence length for each disaster. The results are shown in Table 5 and each entry includes a confusion matrix and four metric values (accuracy, precision, recall, and f1-score). For the Iraq-Iran earthquake disaster, the sequence length of 128 presented the best performance (95.96% accuracy and 95.12% f1-score). Likewise, for Sri Lanka floods, 93.94% accuracy and 85.13% f1-score are the best values with a sequence length of 128. The best results are obtained on the Iraq-Iran earthquake disaster and the lowest results are achieved on the Hurricane Harvey disaster. Moreover, the sequence length of 128 presented the best results for the first two disasters, and for the remaining five disasters, the sequence length of 64 produced the best results.

In the fourth step, we compared the effectiveness of the fine-tuned BERT with state-of-the-art benchmarks (*Alam, Ofli & Imran, 2020*; *Alam et al., 2021*; *Kumar, Singh & Saumya, 2019*; *Madichetty & Sridevi, 2021*; *Rudra et al., 2018*) for damage *vs* non-damage tweets classification. Five benchmarks are compared with a fine-tuned BERT model for each disaster and results are added in Table 6. The fine-tuned BERT outperformed the five benchmarks in all seven disasters. For the Iraq-Iran earthquake, the highest f1-score achieved by the benchmark is 78.48% and the proposed framework demonstrated a 95.12% f1-score. Thus 16.64% improvement is observed in the f1-score. Moreover, fine-tuned BERT demonstrated significant improvement in accuracy, precision, recall, and f1-score in comparison to five benchmarks for seven disasters. Specifically following percentage of improvements are observed in the f1-score; for Hurricane Irma, 5.37%; for

**Table 6 Comparison of fine-tuned BERT classifier with benchmarks (damage *vs* non-damage).**

| Disasters | Model | Accuracy | Macro-precision | Macro-recall | Macro-f1-score |
|---|---|---|---|---|---|
| Iraq–Iran Earthquake | *Rudra et al. (2018)* | 56.67 | 28.33 | 50.00 | 36.00 |
| | *Alam, Ofli & Imran (2020)* | – | 70.04 | 63.40 | 64.8 |
| | *Kumar, Singh & Saumya (2019)* | – | 55.60 | 63.40 | 57.80 |
| | *Alam et al. (2021)* | – | 67.10 | 68.29 | 67.60 |
| | *Madichetty & Sridevi (2021)* | 79.58 | 78.73 | 78.86 | 78.48 |
| | **Proposed** | **95.96** | **96.10** | **94.28** | **95.12** |
| Sri Lanka Floods | *Rudra et al. (2018)* | 50 | 35 | 50 | 40 |
| | *Alam, Ofli & Imran (2020)* | – | 70.04 | 63.4 | 64.8 |
| | *Kumar, Singh & Saumya (2019)* | – | 76.5 | 68 | 67 |
| | *Alam et al. (2021)* | – | 67.1 | 68.29 | 67.6 |
| | *Madichetty & Sridevi (2021)* | 86.92 | 87.26 | 86.92 | 86.89 |
| | **Proposed** | **93.94** | **79.17** | **96.69** | **85.13** |
| Mexico Earthquake | *Rudra et al. (2018)* | 55.83 | 41.67 | 52.5 | 43.33 |
| | *Alam, Ofli & Imran (2020)* | – | 70.04 | 63.4 | 64.8 |
| | *Kumar, Singh & Saumya (2019)* | – | 55.6 | 63.4 | 57.8 |
| | *Alam et al. (2021)* | – | 67.1 | 68.29 | 67.6 |
| | *Madichetty & Sridevi (2021)* | 79.37 | 79.9 | 79.37 | 79.26 |
| | **Proposed** | **92.80** | **88.50** | **89.11** | **88.80** |
| California Wildfires | *Rudra et al. (2018)* | 55.17 | 52.5 | 55.83 | 50.02 |
| | *Kumar, Singh & Saumya (2019)* | – | 44.4 | 46.2 | 44.8 |
| | *Alam et al. (2021)* | – | 67.1 | 68.29 | 67.6 |
| | *Madichetty & Sridevi (2021)* | 73.31 | 73.78 | 73.45 | 73.24 |
| | **Proposed** | **86.26** | **78.24** | **82.58** | **80.03** |
| Hurricane Harvey | *Rudra et al. (2018)* | 45.18 | 29.03 | 43.75 | 33.69 |
| | *Alam, Ofli & Imran (2020)* | – | 70.04 | 63.4 | 64.8 |
| | *Kumar, Singh & Saumya (2019)* | – | 66 | 63.4 | 57.8 |
| | *Alam et al. (2021)* | – | 67.1 | 68.29 | 67.6 |
| | *Madichetty & Sridevi (2021)* | 77.32 | 76.14 | 76.72 | 76.28 |
| | **Proposed** | **89.17** | **77.29** | **80.88** | **78.89** |
| Hurricane Maria | *Rudra et al. (2018)* | 65 | 62.17 | 63.33 | 61.13 |
| | *Alam, Ofli & Imran (2020)* | – | 70.04 | 63.4 | 64.8 |
| | *Kumar, Singh & Saumya (2019)* | – | 66 | 63.4 | 57.8 |
| | *Alam et al. (2021)* | – | 67.1 | 68.29 | 67.6 |
| | *Madichetty & Sridevi (2021)* | 79.14 | 79.41 | 79.14 | 79.1 |
| | **Proposed** | **92.21** | **76.70** | **84.18** | **79.79** |
| Hurricane Irma | *Rudra et al. (2018)* | 48.51 | 36.73 | 48.33 | 40.07 |
| | *Alam, Ofli & Imran (2020)* | – | 70.04 | 63.4 | 64.8 |
| | *Kumar, Singh & Saumya (2019)* | – | 66 | 63.4 | 57.8 |
| | *Alam et al. (2021)* | – | 67.1 | 68.29 | 67.6 |
| | *Madichetty & Sridevi (2021)* | 77.13 | 77.27 | 77.13 | 77.1 |
| | **Proposed** | **91.75** | **82.99** | **81.98** | **82.47** |

Notes.
The bold values are the highest performances achieved by the proposed model for each disaster.

**Table 7 Comparison of fine-tuned BERT with baseline using Wilcoxon's signed-rank test (damage vs non-damage).**

| Disasters | Binary classification | | | |
|---|---|---|---|---|
| | Fine-tuned BERT | Second best | Diff | Rank |
| Iraq–Iran earthquake | 95.12 | 78.48 | 16.64 | 7 |
| Sri Lanka floods | 85.13 | 86.89 | −1.76 | 1 |
| Mexico earthquake | 88.80 | 79.26 | 9.54 | 6 |
| California wildfires | 80.03 | 73.24 | 6.79 | 5 |
| Hurricane Harvey | 78.89 | 76.28 | 2.61 | 3 |
| Hurricane Maria | 79.79 | 79.1 | 0.69 | 2 |
| Hurricane Irma | 82.47 | 77.1 | 5.37 | 4 |

Hurricane Maria, 0.69%; for Hurricane Harvey, 2.61%; for California wildfires, 6.79%; and for Mexico earthquake, 9.54%; This proved the effectiveness of fine-tuned BERT for binary classification of damage assessment tweets and demonstrated better performance than five benchmarks on seven disasters. Among the baselines, the study *Madichetty & Sridevi (2021)* presented better performance than the other four baselines.

In the fifth step, a statistical test is conducted to determine whether the improvements are statistically significant or not. For this, the performance of the fine-tuned BERT model is compared with the best-performing baseline using Wilcoxon's signed-rank test (*Woolson, 2007*) to check the statistical significance of improvements. This test is non-parametric and the null hypothesis can be rejected at the $\alpha$ level. The null hypothesis is that the both models have the same performance. The results of the Wilcoxon signed-rank test are added in Table 7. We compared both models using the macro f1-score for each disaster. The fine-tuned BERT outperforms the baseline for six disasters. The null hypothesis can be rejected on the $\alpha = 0.05$ confidence level. At first, the difference in f1-scores for both models is calculated and the rank is assigned based on absolute difference values. Then, the sum of ranks is calculated following the criteria of adding all positive ranks at one point and adding all negative ranks at another point. We got $R^+ = 7+6+5+3+2+4 = 27$, and $R^- = 1$, where $V_\alpha = 6$. As the minimum sum (*i.e.,* 1) is less than 6, we reject the null hypothesis that both models perform equally. Thus, improvement of fine-tuned BERT is statistically significant for binary classification.

## Fine-tunning of BERT and comparison with baselines (infrastructure *vs* human *vs* non-damage)

In this section, the BERT model is fine-tuned to perform multi-class classification of tweets into infrastructure damage, or human damage, or non-damage categories. After that, the performance of the fine-tuned BERT model is compared with two benchmarks (*Madichetty & Sridevi, 2021*; *Rudra et al., 2018*), and five comparable models.

At first, fine-tuning of BERT is performed using the same parameters described in section 'Fine-tunning of BERT and comparison with baselines (damage vs non-damage)', but here objective is multi-class classification. The 64 and 128 sequence lengths are used with 32 batch size and a learning rate of 2e−5 is used. The outcomes are measured in the

**Table 8 Results of training and validation for several BERT classifiers (multi-class).**

| Sequence length | Batch size | Epochs | Learning rate | Epsilon | Training loss | Validation loss | Validity accuracy | Validating F1-score | Training time | Validation time |
|---|---|---|---|---|---|---|---|---|---|---|
| BERT -64 | 32 | 1 | 2e−5 | 1e−8 | 0.35 | 0.29 | 0.8948 | 0.8422 | 0:02:03 | 0:00:05 |
| | | 2 | 2e−5 | 1e−8 | 0.21 | 0.25 | 0.9083 | 0.8578 | 0:01:59 | 0:00:04 |
| | | 3 | 2e−5 | 1e−8 | 0.13 | 0.35 | 0.9083 | 0.8578 | 0:01:59 | 0:00:05 |
| | | 4 | 2e−5 | 1e−8 | 0.09 | 0.39 | 0.9083 | 0.8578 | 0:01:59 | 0:00:05 |
| BERT -128 | 32 | 1 | 2e−5 | 1e−8 | 0.35 | 0.27 | 0.8948 | 0.8291 | 0:03:42 | 0:00:08 |
| | | 2 | 2e−5 | 1e−8 | 0.21 | 0.25 | 0.9083 | 0.8844 | 0:03:40 | 0:00:08 |
| | | 3 | 2e−5 | 1e−8 | 0.14 | 0.36 | 0.8948 | 0.8291 | 0:03:39 | 0:00:08 |
| | | 4 | 2e−5 | 1e−8 | 0.09 | 0.38 | 0.8948 | 0.8291 | 0:03:38 | 0:00:08 |

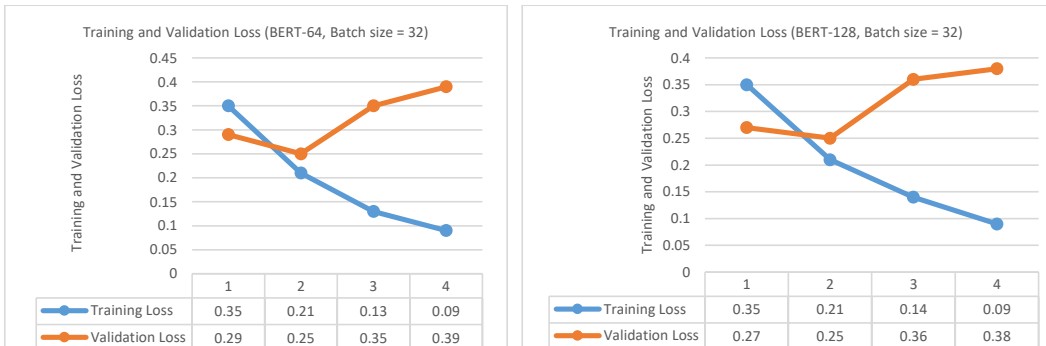

**Figure 5 Training loss and Validation loss (infrastructure *vs* human *vs* non-damage) (for sequence length of 64 and batch size of 32 (left)), (for sequence length of 128 and batch size of 32 (right)).**

form of training and validation loss, validation accuracy, and validation f1-score. 80% of all seven disaster data is used for training purposes and the remaining 20% is used for testing purposes. Of 80%, 90% is actually used for training and 10% is used for validation. The training and validation results of fine-tuning BERT are presented in Table 8. For a sequence length of 64, the best validation accuracy is 90.83% achieved on epoch 2. Likewise, the best validation f1-score is obtained on epoch 2, *i.e.*, 85.78%. The response of training loss is continuously decreasing indicating that the model is learning steadily during training as shown in Fig. 5 (left side). The validation loss decreases from epochs 1 to 2 and then starts increasing up to epoch 4, representing that further training may lead to overfitting. The lower part of Table 8 shows the results generated using a sequence length of 128. The response of training loss is again decreasing from epoch 1-4, in contrast, validation loss decreases on epoch 2 but then starts increasing up to epoch 4 as shown in Fig. 5 (right side). The highest values obtained for validation accuracy and f1-score are 90.83% and 88.44% on epoch 2. The four BERT classifiers are then tested on the test parts of seven

disasters. This completes the training and validation of fine-tuning of BERT classifiers for multi-class classification.

After that, we tested the BERT classifiers on the test part of each disaster and the best results are reported in Tables 9 and 10. In addition, the performance of the fine-tuned BERT model is compared with two benchmarks (*Madichetty & Sridevi, 2021*; *Rudra et al., 2018*), and five comparable models. There are nine comparable models but we added results of the top five comparable models. The models are trained on 80% data and then tested on the remaining 20% of the disaster datasets. Table 9 shows the accuracy and per class (infrastructure *vs* human *vs* non-damage) f1-score values with macro-average. For the Iraq-Iran earthquake disaster, the fine-tuned BERT outperformed and obtained accuracy (93.88%), precision (85.40%), recall (80.66%), and macro-f1-score (82.63%). The improvement of 6.13% in macro-f1 is obtained by the proposed framework as compared to the benchmark (*Madichetty & Sridevi, 2021*). Moreover, we can notice the improvement in human damage and non-damage classification presented by the proposed framework as compared to benchmarks and five comparable models. An improvement of 16.20% in human damage and 19.37% in non-damage category classification is observed. Likewise, for macro-precision and macro-recall metrics, it is evident from Table 10 that 4.5% and 5.79% improvement is achieved by the proposed framework compared to the benchmark (*Madichetty & Sridevi, 2021*).

For the Sri Lanka flood disaster, the proposed framework outperformed the two benchmarks, and five comparable models by demonstrating a 10.33% improvement in macro f1-score, 12.28% in accuracy, 0.53% in macro-precision, and 17.89% in macro-recall measures. Moreover, improvement of 1.25% in infrastructure damage, 17.2% in human damage, and 12.53% in non-damage are observed in the f1-score. For the Mexico earthquake disaster, the proposed framework outperformed all models. An improvement of 6.64% in macro f1-score, 9.2% in macro-precision, and 4.87% in macro-recall are detected compared to benchmarks. Considering the California wildfire disaster, the fine-tuned BERT model delivered the highest performance. We observe an improvement in macro f1-score (12.95%), macro-precision (1.25%), and macro-recall (14.74%). Furthermore, an improvement of 13.16% in human damage, and 26.98% in non-damage classification is obtained by the fine-tuned BERT model.

For the Hurricane Harvey disaster, the proposed framework presented much better performance than state-of-the-art benchmarks and five comparable models. Considering the f1-score, the proposed framework obtained 8.54%, 19.17%, and 6.94% improvements in human damage, non-damage and macro f1-score. Furthermore, 2.33% and 10.89% improvements are observed in macro precision and macro recall. For the Hurricane Maria and Hurricane Irma disasters, the proposed framework performed better than benchmarks and comparable models in accuracy as shown in Table 9. Regarding the f1-score, non-damage classification is improved by 17.7% and 21.07% for Hurricane Maria and Hurricane Irma but did not perform better in the macro f1-score. This shows that fine-tuned BERT is not trained well on these two disaster datasets. This completes the evaluation of the proposed framework on seven disasters for binary classification and multi-class classification.

**Table 9  Comparison of fine-tuned BERT with benchmarks for multi-class classification (accuracy and f1-score).**

| Disasters | Model | Accuracy | F1-score | | | |
|---|---|---|---|---|---|---|
| | | | Infrastructure | Human | Non | Macro |
| Iraq–Iran Earthquake | *Rudra et al. (2018)* | – | 0 | 69.36 | 65.13 | 44.83 |
| | Word2vec+RF | 74.00 | 45.71 | 84.06 | 59.26 | 63.01 |
| | TF-IDF+SVM | 86.87 | 28.57 | 76.19 | 92.62 | 65.79 |
| | TF-IDF+LSTM | 87.79 | 48.89 | 84.55 | 92.65 | 75.36 |
| | TF-IDF+Bi-LSTM | 88.27 | 50.19 | 85.34 | 92.70 | 76.07 |
| | TF-IDF+CNN | 89.29 | 53.10 | 86.28 | 92.78 | 77.38 |
| | *Madichetty & Sridevi (2021)* | – | 74.33 | 80.00 | 75.18 | 76.50 |
| | **Proposed** | **93.88** | **57.14** | **96.20** | **94.55** | **82.63** |
| Sri Lanka Floods | *Rudra et al. (2018)* | – | 41.72 | 64.56 | 39.52 | 48.60 |
| | Word2vec+RF | 74.00 | 45.71 | 84.06 | 59.26 | 63.01 |
| | TF-IDF+RF | 81.55 | 33.33 | 68.29 | 88.89 | 63.50 |
| | TF-IDF+LSTM | 82.32 | 47.64 | 84.18 | 89.71 | 73.84 |
| | TF-IDF+Bi-LSTM | 83.53 | 49.71 | 84.47 | 90.32 | 74.83 |
| | TF-IDF+CNN | 84.10 | 51.43 | 84.90 | 91.54 | 75.95 |
| | *Madichetty & Sridevi (2021)* | – | 74.61 | 74.69 | 83.59 | 77.63 |
| | **Proposed** | **93.83** | **75.86** | **91.89** | **96.12** | **87.96** |
| Mexico Earthquake | *Rudra et al. (2018)* | – | 0 | 63.33 | 63.31 | 42.21 |
| | Word2vec+RF | 74.00 | 45.71 | 84.06 | 59.26 | 63.01 |
| | TF-IDF+SVM | 91.43 | 53.85 | 78.43 | 95.40 | 75.89 |
| | TF-IDF+LSTM | 91.20 | 54.18 | 79.10 | 86.39 | 73.22 |
| | TF-IDF+Bi-LSTM | 91.30 | 54.58 | 79.40 | 87.10 | 73.69 |
| | TF-IDF+CNN | 91.33 | 55.38 | 79.90 | 87.37 | 74.21 |
| | *Madichetty & Sridevi (2021)* | – | 61.15 | 71.21 | 70.65 | 67.67 |
| | **Proposed** | **91.39** | **57.14** | **70.27** | **95.51** | **74.31** |
| California Wildfires | *Rudra et al. (2018)* | – | 39.81 | 51.35 | 53.18 | 48.11 |
| | Word2vec+RF | 74.00 | 45.71 | 84.06 | 59.26 | 63.01 |
| | TF-IDF+SVM | 85.42 | 26.32 | 73.33 | 91.46 | 63.70 |
| | TF-IDF+LSTM | 83.01 | 45.82 | 83.27 | 88.10 | 72.39 |
| | TF-IDF+Bi-LSTM | 83.12 | 46.29 | 83.43 | 88.21 | 72.64 |
| | TF-IDF+CNN | 83.27 | 46.76 | 83.64 | 88.38 | 72.92 |
| | *Madichetty & Sridevi (2021)* | – | 55.02 | 68.98 | 61.53 | 61.84 |
| | **Proposed** | **83.33** | **53.73** | **82.14** | **88.51** | **74.79** |
| Hurricane Harvey | *Rudra et al. (2018)* | – | 53.59 | 13.01 | 43.97 | 36.86 |
| | Word2vec+RF | 74.00 | 45.71 | 84.06 | 59.26 | 63.01 |
| | TF-IDF+SVM | 87.88 | 39.58 | 23.53 | 93.39 | 52.17 |
| | TF-IDF+LSTM | 85.58 | 55.37 | 64.29 | 91.87 | 70.51 |
| | TF-IDF+Bi-LSTM | 85.67 | 56.48 | 64.59 | 92.12 | 71.06 |
| | TF-IDF+CNN | 85.87 | 57.84 | 64.7 | 92.43 | 71.65 |
| | *Madichetty & Sridevi (2021)* | – | 68.52 | 58.69 | 72.54 | 66.58 |
| | **Proposed** | **86.11** | **61.62** | **67.23** | **91.71** | **73.52** |

**Table 9** (*continued*)

| Disasters | Model | Accuracy | F1-score | | | |
|---|---|---|---|---|---|---|
| | | | **Infrastructure** | **Human** | **Non** | **Macro** |
| | *Rudra et al. (2018)* | – | 0 | 77.36 | 77.87 | 51.74 |
| | TF-IDF+SVM | 91.70 | 23.73 | 22.22 | 95.61 | 47.19 |
| | Word2vec+RF | 73.79 | 48.65 | 84.29 | 55.17 | 62.70 |
| Hurricane Maria | TF-IDF+LSTM | 91.79 | 49.58 | 43.32 | 95.87 | 62.92 |
| | TF-IDF+Bi-LSTM | 91.87 | 49.86 | 44.13 | 96.09 | 63.36 |
| | TF-IDF+CNN | 92.10 | 50.38 | 46.67 | 96.35 | 64.46 |
| | *Madichetty & Sridevi (2021)* | – | 75.92 | 57.13 | 79.14 | 70.73 |
| | **Proposed** | **93.84** | **50.7** | **52.94** | **96.84** | **66.83** |
| | *Rudra et al. (2018)* | – | 59.5 | 0 | 54.08 | 37.86 |
| | TF-IDF+SVM | 87.42 | 23.91 | 19.05 | 93.41 | 45.46 |
| | Word2vec+RF | 71.43 | 47.37 | 82.27 | 51.61 | 60.42 |
| Hurricane Irma | TF-IDF+LSTM | 88.58 | 29.7 | 40.58 | 93.59 | 54.62 |
| | TF-IDF+Bi-LSTM | 88.87 | 30.49 | 40.78 | 93.7 | 54.99 |
| | TF-IDF+CNN | 89.29 | 31.89 | 41.28 | 93.8 | 55.65 |
| | *Madichetty & Sridevi (2021)* | – | 75.26 | 49.89 | 74.34 | 66.50 |
| | **Proposed** | **91.24** | **56.6** | **18.75** | **95.41** | **56.92** |

**Notes.**
The bold values are the highest performances achieved by the proposed model for each disaster.

In the end, the statistical test is performed using the Wilcoxon signed-ranked test to check whether the improvements of the proposed model are statistically significant or not for multi-class classification. As the Wilcoxon sined-ranked test applies to two classifiers, therefore fine-tuned BERT model is compared with the second-best performing model, and the results are reported in Table 11. For justifying the improvements to be statistically significant, our analysis should reject the null hypothesis. The performance is compared in macro f1-score for each disaster and fine-tuned BERT outperformed the second-best for five disasters. After calculating the difference in performance, the ranks are assigned using absolute values. Then the sum of ranks is calculated and we got $R^+ = 25$ and $R^- = 3$, where $V_\alpha = 5$. By comparing $R^-$ and $V_\alpha$, the former is less than the latter and we reject the null hypothesis. Hence, the improvements of the fine-tuned BERT model are statistically significant for multi-class classification.

## DISCUSSION AND LIMITATIONS

Assessment of damages and proper coordination of rescue efforts are in high demand for in-time response to disasters. Recently, the emergence of state-of-the-art deep learning technologies attracted the researchers' attention, and robust damage identification models can be developed using these DL techniques and taking benefits of available big datasets. However, the lack of comprehension of the strengths and limitations of these technologies, especially in comparison with traditional ML techniques and deployment issues, requires further investigation. In this research, we propose a tool for damage assessment from textual data, based upon a benchmark BERT transformer model with fine-tuning. This tool supports two levels of identification of damages from tweets: binary and multi-class

**Table 10  Comparison of fine-tuned BERT with benchmarks for multi-class classification (precision and recall).**

| Disasters | Model | Precision | | | | Recall | | | |
|---|---|---|---|---|---|---|---|---|---|
| | | Infra | Human | Non | Macro | Infra | Human | Non | Macro |
| | *Rudra et al. (2018)* | 0 | 74.53 | 56.99 | 43.84 | 0 | 66.92 | 77.73 | 48.22 |
| | Word2vec+RF | 57.14 | 76.32 | 80.00 | 71.15 | 38.10 | 93.55 | 47.06 | 59.57 |
| | TF-IDF+SVM | 99.0 | 88.89 | 86.25 | 91.71 | 16.67 | 66.67 | 99.0 | 60.78 |
| Iraq–Iran | TF-IDF+LSTM | 79.12 | 78.69 | 87.64 | 81.81 | 40.89 | 84.79 | 92.55 | 72.74 |
| Earthquake | TF-IDF+Bi-LSTM | 82.38 | 81.21 | 88.24 | 83.94 | 43.19 | 85.58 | 92.60 | 73.79 |
| | TF-IDF+CNN | 83.15 | 82.76 | 89.74 | 85.21 | 45.10 | 86.83 | 92.68 | 74.86 |
| | *Madichetty & Sridevi (2021)* | 88.33 | 79.88 | 74.50 | 80.90 | 66.67 | 80.66 | 77.27 | 74.87 |
| | **Proposed** | **66.67** | **95.00** | **94.55** | **85.40** | **50.00** | **97.44** | **94.55** | **80.66** |
| | *Rudra et al. (2018)* | 42.08 | 63.33 | 64.94 | 56.78 | 44.50 | 70.50 | 37.50 | 50.83 |
| | Word2vec+RF | 57.14 | 76.32 | 80.00 | 71.15 | 38.10 | 93.55 | 47.06 | 59.57 |
| | TF-IDF+RF | 50.00 | 87.50 | 81.93 | 73.14 | 25.00 | 56.00 | 97.14 | 59.38 |
| Sri Lanka | TF-IDF+LSTM | 58.45 | 87.90 | 82.69 | 76.34 | 41.49 | 84.10 | 88.34 | 71.31 |
| Floods | TF-IDF+Bi-LSTM | 59.78 | 88.24 | 83.87 | 77.29 | 43.58 | 84.58 | 89.46 | 72.54 |
| | TF-IDF+CNN | 60.42 | 88.69 | 84.79 | 77.96 | 45.69 | 85.69 | 90.67 | 74.01 |
| | *Madichetty & Sridevi (2021)* | 85.33 | 84.83 | 78.02 | 82.73 | 70.50 | 70.50 | 93.00 | 78.0 |
| | **Proposed** | **61.11** | **89.47** | **99.20** | **83.26** | **100.00** | **94.44** | **93.23** | **95.89** |
| | *Rudra et al. (2018)* | 50.59 | 69.00 | 50.32 | 39.77 | 0 | 59.61 | 85.98 | 48.53 |
| | TF-IDF+SVM | 87.50 | 90.91 | 91.63 | 90.01 | 38.89 | 68.97 | 99.49 | 69.12 |
| | Word2vec+RF | 57.14 | 76.32 | 80.00 | 71.15 | 38.10 | 93.55 | 47.06 | 59.57 |
| Mexico | TF-IDF+LSTM | 59.38 | 77.71 | 81.49 | 72.86 | 39.56 | 93.78 | 47.43 | 60.25 |
| Earthquake | TF-IDF+Bi-LSTM | 59.85 | 78.32 | 82.38 | 73.51 | 40.32 | 93.97 | 49.39 | 61.22 |
| | TF-IDF+CNN | 60.12 | 79.78 | 83.58 | 74.49 | 41.58 | 94.12 | 51.78 | 62.49 |
| | *Madichetty & Sridevi (2021)* | 66.02 | 73.05 | 67.71 | 68.93 | 58.27 | 70.16 | 74.25 | 67.56 |
| | **Proposed** | **53.33** | **86.67** | **94.39** | **78.13** | **61.54** | **59.09** | **96.65** | **72.43** |
| | *Rudra et al. (2018)* | 33.15 | 54.16 | 72.78 | 53.36 | 51.00 | 50.32 | 43.76 | 48.36 |
| | TF-IDF+SVM | 71.43 | 100.0 | 84.59 | 85.34 | 16.13 | 57.89 | 99.56 | 57.86 |
| | Word2vec+RF | 57.14 | 76.32 | 80.00 | 71.15 | 38.10 | 93.55 | 47.06 | 59.57 |
| California | TF-IDF+LSTM | 72.59 | 99.10 | 85.19 | 85.62 | 51.29 | 70.45 | 82.87 | 68.20 |
| Wildfires | TF-IDF+Bi-LSTM | 73.39 | 99.31 | 86.49 | 86.39 | 51.78 | 73.59 | 83.29 | 69.55 |
| | TF-IDF+CNN | 74.19 | 99.43 | 87.69 | 87.10 | 52.10 | 75.98 | 85.48 | 71.18 |
| | *Madichetty & Sridevi (2021)* | 59.98 | 72.14 | 57.67 | 63.26 | 51.69 | 66.87 | 66.95 | 61.84 |
| | **Proposed** | **54.55** | **75.41** | **90.50** | **73.49** | **52.94** | **90.20** | **86.60** | **76.58** |
| | *Rudra et al. (2018)* | 42.42 | 46.22 | 49.98 | 46.21 | 73.23 | 7.92 | 40.06 | 40.40 |
| | TF-IDF+SVM | 65.52 | 85.71 | 88.76 | 80.00 | 28.36 | 13.64 | 98.53 | 46.84 |
| | Word2vec+RF | 57.14 | 76.32 | 80.00 | 71.15 | 38.10 | 93.55 | 47.06 | 59.57 |
| Hurricane | TF-IDF+LSTM | 66.59 | 76.57 | 81.23 | 74.79 | 39.49 | 93.71 | 48.39 | 60.53 |
| Harvey | TF-IDF+Bi-LSTM | 66.74 | 76.75 | 81.46 | 74.98 | 40.36 | 93.85 | 50.25 | 61.48 |
| | TF-IDF+CNN | 67.21 | 77.04 | 81.76 | 75.33 | 41.59 | 94.10 | 51.59 | 62.42 |
| | *Madichetty & Sridevi (2021)* | 73.34 | 64.31 | 67.17 | 68.27 | 64.97 | 55.25 | 79.55 | 66.59 |
| | **Proposed** | **53.98** | **63.49** | **94.32** | **70.60** | **71.76** | **71.43** | **89.25** | **77.48** |
| | *Rudra et al. (2018)* | 45.15 | 0 | 53.63 | 32.97 | 76.90 | 0 | 40.34 | 39.08 |
| | Word2vec+RF | 60.00 | 76.62 | 72.73 | 69.78 | 40.91 | 93.65 | 44.44 | 59.67 |

**Table 10** (*continued*)

| Disasters | Model | Precision | | | | Recall | | | |
|---|---|---|---|---|---|---|---|---|---|
| | | **Infra** | **Human** | **Non** | **Macro** | **Infra** | **Human** | **Non** | **Macro** |
| Hurricane Maria | TF-IDF+SVM | 77.78 | 100.0 | 91.83 | 89.87 | 14.00 | 12.50 | 99.72 | 42.07 |
| | TF-IDF+LSTM | 78.10 | 98.21 | 91.95 | 89.42 | 18.45 | 20.38 | 90.10 | 42.97 |
| | TF-IDF+Bi-LSTM | 78.48 | 98.43 | 92.13 | 89.68 | 20.27 | 22.87 | 90.56 | 44.56 |
| | TF-IDF+CNN | 78.69 | 98.65 | 92.37 | 89.90 | 32.78 | 25.76 | 92.18 | 50.24 |
| | *Madichetty & Sridevi (2021)* | 77.16 | 71.17 | 74.73 | 74.35 | 75.52 | 48.63 | 85.52 | 69.89 |
| | **Proposed** | **47.37** | **42.86** | **97.69** | **62.64** | **54.55** | **69.23** | **95.99** | **73.26** |
| Hurricane Irma | *Rudra et al. (2018)* | 50.17 | 0 | 57.10 | 35.76 | 73.95 | 0 | 52.47 | 42.14 |
| | TF-IDF+SVM | 61.11 | 57.14 | 88.31 | 68.86 | 14.86 | 11.43 | 99.13 | 41.81 |
| | Word2vec+RF | 56.25 | 76.32 | 61.54 | 64.70 | 40.91 | 89.23 | 44.44 | 58.19 |
| | TF-IDF+LSTM | 61.49 | 77.38 | 88.53 | 75.80 | 42.41 | 88.49 | 44.78 | 58.56 |
| | TF-IDF+Bi-LSTM | 62.21 | 77.79 | 89.11 | 76.37 | 43.89 | 89.34 | 45.29 | 59.50 |
| | TF-IDF+CNN | 62.54 | 78.10 | 89.34 | 76.66 | 44.11 | 89.67 | 45.54 | 59.77 |
| | *Madichetty & Sridevi (2021)* | 75.24 | 59.81 | 71.79 | 68.95 | 76.19 | 43.42 | 77.92 | 65.84 |
| | **Proposed** | **48.39** | **20.00** | **96.48** | **54.96** | **68.18** | **17.65** | **94.36** | **60.06** |

**Notes.**
The bold values are the highest performances achieved by the proposed model for each disaster.

**Table 11** **Verification of improvements using Wilcoxon's signed-rank test (Multi-class).**

| Dataset | Multi-class Classification | | | |
|---|---|---|---|---|
| | **Fine-tuned BERT** | **Second Best** | **Diff** | **Rank** |
| Iraq–Iran Earthquake | 82.63 | 77.38 | 5.25 | 6 |
| Sri Lanka Floods | 87.96 | 77.63 | 10.33 | 7 |
| Mexico Earthquake | 74.31 | 74.21 | 0.1 | 3 |
| California Wildfires | 74.78 | 72.92 | 1.86 | 4 |
| Hurricane Harvey | 73.52 | 71.65 | 1.87 | 5 |
| Hurricane Maria | 66.83 | 70.73 | −3.9 | 2 |
| Hurricane Irma | 56.92 | 66.50 | −9.58 | 1 |

classification. One effective application of this tool in rehabilitation and rescue stages would be the utilization of quantitative statistics of damages with some qualitative approaches, to verify how much damages are and how these influence individuals and societies.

This study made significant contributions in the domain of damage tweet identification and crisis management in real-time disasters. The fine-tuning of BERT transformer model is studied first time for binary as well as multi-class classification according to our knowledge. The most important contribution is the identification of human and infrastructure damages at the second level. The utilization of contextual semantic embeddings enables us to handle the ambiguity and complexity issues of language used to describe damages. The optimization of hyper-parameters for the fine-tuning process helped us to handle the issues of overfitting and catastrophic forgetting. For binary classification, it outperformed the five benchmarks for all disaster datasets and demonstrated significant improvement in detecting damage assessment tweets. For multi-class classification, it outperformed the two benchmarks, and all comparable models for five disaster datasets and presented the

comparable performance for the remaining two disasters. This proves the effectiveness of fine-tuned BERT for damage assessment tweet identification both for binary and multi-class classification.

In the end, the advantages of the proposed framework are summarized as follows: First, the baselines are developed using hand-crafted features but the proposed framework utilized an automated feature generation model to address the issue of damage assessment tweets identification. Second, the language model has the ability to capture the actual context of the language being used (to describe the infrastructure and human damages) in the tweets instead of semantic, syntactic, and frequency-based approaches. Third, on top of everything, the performance improvement achieved by the proposed framework on seven disaster datasets compared to benchmarks is promising.

There are some limitations of this study. Our dataset that contains seven disasters is limited to 18,084 tweets and this dataset (CrisisMMD) covers only seven real-time disasters. The size and comprehensiveness of the dataset can be addressed as a potential avenue for improvement. As our methodology is based on a deep learning paradigm, therefore more larger and comprehensive dataset would definitely improve the performance of identification. Another limitation is that although the proposed model addresses multi-class categorization of tweets, it is still not able to assess the number of damages related to the human and infrastructure categories. Such kind of assessments will be very helpful for rehabilitation organizations to estimate the losses and damages before arrival at disaster locations. Future work can extend the findings of this study to propose a solution for the assessment of the quantity of damages.

## CONCLUSION

This research investigated the issue of damage assessment tweet identification as a binary and multi-class classification mechanism. We proposed a "contextual semantic embeddings" based model for improving damage assessment tweet identification. To handle the issues of complexity and ambiguity and to support the generalization of the framework, a state-of-the-art language model (BERT) is utilized with fine-tuning important hyper-parameters without relying on basic and hand-crafted features. Moreover, nine comparable models are designed to compare the performance of the proposed framework. Several BERT classifiers are trained, validated, and tested by fine-tuning important hyper-parameters. To evaluate the effectiveness of the proposed framework, the CrisisMMD dataset containing seven disasters is considered. The results demonstrated that the fine-tuned BERT model outperformed all benchmarks and comparable models for binary classification as well as for multi-class classification. Specifically, the proposed framework demonstrated state-of-the-art results by obtaining a macro f1-score of 95.12% for binary classification and a macro f1-score of 88% for multi-class classification. The findings of our study would help disaster management and humanitarian organizations to better manage their rescue activities on time.

In the future, we are interested in exploring fine-tuning language models with adapter mechanisms for similar NLP tasks to reduce the training parameter complexities. The

accuracy of results for damage assessment tweet identification could be improved by utilizing some state-of-the-art hybrid methodologies. Furthermore, the explainability of damage assessment tweet identification will be very helpful for rescue and disaster management organizations to manage their services on time.

## ACKNOWLEDGEMENTS

This article is an output of a research project implemented as part of the Basic Research Program at the National Research University Higher School of Economics (HSE University). This research made use of computational resources of HPC facilities at HSE University.

### Funding

This study was supported by Princess Nourah bint Abdulrahman University Researchers Supporting Project number (PNURSP2024R104), Princess Nourah bint Abdulrahman University, Riyadh, Saudi Arabia. The funders had no role in study design, data collection and analysis, decision to publish, or preparation of the manuscript.

### Grant Disclosures

The following grant information was disclosed by the authors:
Princess Nourah bint Abdulrahman University, Riyadh, Saudi Arabia: PNURSP2024R104.

### Competing Interests

The authors declare there are no competing interests.

### Author Contributions

- Muhammad Shahid Iqbal Malik conceived and designed the experiments, performed the experiments, analyzed the data, performed the computation work, prepared figures and/or tables, authored or reviewed drafts of the article, and approved the final draft.
- Muhammad Zeeshan Younas conceived and designed the experiments, performed the experiments, analyzed the data, performed the computation work, prepared figures and/or tables, and approved the final draft.
- Mona Mamdouh Jamjoom performed the experiments, authored or reviewed drafts of the article, and approved the final draft.
- Dmitry I. Ignatov performed the experiments, authored or reviewed drafts of the article, and approved the final draft.

### Data Availability

The data is available at CrisisNLP: https://crisisnlp.qcri.org/crisismmd.
The code is available in the Supplemental Files.

## Supplemental Information

Supplemental information for this article can be found online at http://dx.doi.org/10.7717/peerj-cs.1859#supplemental-information.

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
