# Peer review of "Categorization of tweets for damages: infrastructure and human damage assessment using fine-tuned BERT model"

_PeerJ Computer Science, doi:10.7717/peerj-cs.1859_

## Round 0.1 · original submission · Major Revisions

Based on the reviewers' suggestions and my opinion, I suggest the paper needs major revisions.

**Language Note:** The review process has identified that the English language must be improved. PeerJ can provide language editing services - please contact us at [email protected] for pricing (be sure to provide your manuscript number and title). Alternatively, you should make your own arrangements to improve the language quality and provide details in your response letter. – PeerJ Staff

·

Basic reporting

1. What is the novelty of this paper? Clearly mention it in the introduction section.
2. The proposed algorithm and flow of instructions are missing you should add it in your manuscript.
3. The discussion section is missing you should add it to your manuscript

Experimental design

1. In Figure 1 what is the use of output of training from SVM, LR, and RF in test data.
2. What is the contribution of the ML classifier over the BERT classifier?
3. Clearly explain Figure 1.

Validity of the findings

1. Result section is week. You should add more model results such as DNN, CNN, LSTM, Bi-LSTM, LSTM-A, etc., for comparison of your proposed model.
2. The time complexity of your proposed model should be compared with the existing state-of-the-art models.

Additional comments

1. At line no. 82, what is nn-damage?
2. Lot of work have already been done on this field. Without citing then you work seems to be incomplete for example you may refer to the work done by

Reviewer 2 ·

Basic reporting

The authors need to consider the following points:


1. Refine the paper for better flow and viewpoint.

2. Some bad English constructions and misuse of articles. Therefore, a professional language editing service is strongly recommended.

3. More description of the technical details will help to improve the quality of the manuscript.

4. It is unclear how the different steps in the proposed model are implemented. The steps would be better explained with an example.

5. Different techniques and algorithms are used. However, there is no a clear justification for why these techniques and algorithms should be used rather than others.

6. There are no statistically significant results for the improvement. A statistical test should be conducted to determine whether any improvements are statistically significant.

7. Report clearly the limitations of this work.

Experimental design

No comment

Validity of the findings

No comment

Additional comments

No comment

Reviewer 3 ·

Basic reporting

There are few typos.
Line 82 nn-damage is likely to be a typo, please fix.
Line 178–183, please be consistent with syntax. For example in line 180 replace -> replacement of
Line 236 reached on the conclusion -> reached the conclusion
Line 239: Determining the number of epochs in the fine-tuning stage is a common issue is training the weights of deep learning models.

Experimental design

In the experiment section, the authors compared three models -- pre-trained BERT, word2Vec, and TFIDF on a known benchmark datasetCrisisMMD. The dataset is also well-introduced.

Validity of the findings

The paper mostly compared existing models and fine-tuned the BERT model on a specific dataset. The results are valid but more novelty would be appreciated.

---

## Round 0.2 · accepted · Accept

The paper has addressed all reviewers' questions.

·

Basic reporting

All ok

Experimental design

Done as per the suggestions

Validity of the findings

Satisfied

Additional comments

None